# Paulownia Witches’ Broom Disease: A Comprehensive Review

**DOI:** 10.3390/microorganisms12050885

**Published:** 2024-04-28

**Authors:** Yajie Zhang, Zesen Qiao, Jidong Li, Assunta Bertaccini

**Affiliations:** 1College of Forestry, Henan Agricultural University, Zhengzhou 450000, China; zyjxww@163.com (Y.Z.); zesenqiao@163.com (Z.Q.); 2Henan Provincial Institute of Scientific and Technical Information, Zhengzhou 450003, China; 3Department of Agriculture and Food Science, *Alma Mater Studiorum*—University of Bologna, 40127 Bologna, Italy

**Keywords:** plant disease, phytoplasmas, epidemiology

## Abstract

Phytoplasmas are insect-transmitted bacterial pathogens associated with diseases in a wide range of host plants, resulting in significant economic and ecological losses. Perennial deciduous trees in the genus *Paulownia* are widely planted for wood harvesting and ornamental purposes. Paulownia witches’ broom (PaWB) disease, associated with a 16SrI-D subgroup phytoplasma, is a destructive disease of paulownia in East Asia. The PaWB phytoplasmas are mainly transmitted by insect vectors in the Pentatomidae (stink bugs), Miridae (mirid bugs) and Cicadellidae (leafhoppers) families. Diseased trees show typical symptoms, such as branch and shoot proliferation, which together are referred to as witches’ broom. The phytoplasma presence affects the physiological and anatomical structures of paulownia. Gene expression in paulownia responding to phytoplasma presence have been studied at the transcriptional, post-transcriptional, translational and post-translational levels by high throughput sequencing techniques. A PaWB pathogenic mechanism frame diagram on molecular level is summarized. Studies on the interactions among the phytoplasma, the insect vectors and the plant host, including the mechanisms underlying how paulownia effectors modify processes of gene expression, will lead to a deeper understanding of the pathogenic mechanisms and to the development of efficient control measures.

## 1. Introduction

Phytoplasmas are wall-less bacterial plant pathogens provisionally classified to the ‘*Candidatus* Phytoplasma’ genus [1,2]; they are mainly transmitted by insect vectors and inhabit the phloem of plants and the hemolymphs of insects. After colonization, phytoplasmas secrete effector proteins into cytoplasm of the host plant cells. These effectors interact with certain target proteins in the plant cell, manipulate several host metabolic pathways inducing a series of disease symptoms that includes witches’ broom (shoot proliferation), phyllody (leafy flower) and virescence, leaf yellowing and decline, stunted and little leaf, white leaf, purple top and other malformations. Some of the symptoms, such as witches’ broom and phyllody, increase the prevalence of short branches and small young leaves, enhancing attraction of insect vectors and thus benefit the spread of phytoplasmas [1,2].

The axenic culture of phytoplasmas is difficult to achieve; thus, the Koch postulates to confirm their role as pathogens are not yet fulfilled. Limited knowledge of their biological properties hindered their classification; therefore, phytoplasmas were classified using 16S rRNA gene sequences. Up to now, there are 36 published 16Sr phytoplasma groups and 49 ‘*Ca*. Phytoplasma’ species that infect more than 1000 plant species and cause significant economic losses [3,4]. 

The genus *Paulownia*, within the Paulowniaceae family, includes 17 fast growing, hard wood tree species. The most commonly cultivated species are *Paulownia fortunei*, *P. catalpifolia*, *P. tomentosa* and *P. elongate* [5,6]. The paulownias are mainly distributed in south-eastern Asia (especially China), where they were cultivated for over 2000 years for wood and ornamental purposes [6]. The genus name *Paulownia* Siebold & Zucc was recorded by German botanist Philipp Franz von Siebold, to honor Anna Pavlovna Romanova, grand duchess of Russia and queen of the Netherlands, who sponsored Philipp’s expedition to Japan. Therefore, the paulownia tree is also known as the princess tree and the royal tree [7], while its Chinese name is 泡桐 (pao tong). Nowadays, paulownia trees are widely planted across Asia, North America, Europe and Australia for commercial and decorative purposes [6]. 

Paulownia witches’ broom (PaWB) (Figure 1) is the most destructive disease affecting paulownia and causes serious economic losses [8]. It is associated with the presence of PaWB phytoplasmas [9] and it has been studied for more than 100 years; however, the pathogenic mechanisms, especially the molecular mechanisms underlying the disease symptoms, are still not clear. Efficient and economically feasible prevention and control methods for PaWB have still not been achieved. This review summarizes both the past of PaWB disease research, and the up-to-date knowledge about research on the pathogenic mechanisms of the associated phytoplasma at the molecular level. A PaWB pathogenic mechanism frame diagram on molecular level is summarized in Figure 2, and future and prospective research directions are put forward. 

## 2. Historical Background

PaWB was one of the first phytoplasma diseases in the world to be studied. According to Kawakami’s report in 1902, PaWB disease was first observed in 1880, in Kumamoto, Japan [10]. In China, the Japanese phytopathologist, Ichiro Miyake, serving as a professor in the Imperial University of Peking, now China Agricultural University, reported “paulownia anthrax disease”, with witches’ broom symptoms, in Liaoning province in 1910 [11]. An occurrence of PaWB disease was also reported in Korea [12]. 

Since PaWB could be transmitted by grafting, it was first believed to be a viral disease. In 1967, Doi et al. discovered mycoplasma-like structures in ultrathin sections of the phloem of plants affected by paulownia witches’ broom, as well as mulberry dwarf and aster yellows [13]. The term mycoplasma-like organisms (MLOs) was used to refer to the agents associated with these diseases. In 1992, the name “phytoplasma” was adopted by the Phytoplasma Working Team at the 9th Congress of International Organization of Mycoplasmology to collectively denote MLOs [14]. In 2004, these non-helical, cell wall-less bacteria were placed within the novel provisional genus ‘*Candidatus* Phytoplasma’ [15]. The PaWB phytoplasma is a strain of ‘*Candidatus* Phytoplasma asteris’, belonging to ribosomal subgroup 16SrI-D [3,4,9]. 

## 3. The Paulownia Witches’ Broom Phytoplasma

Although the cultivation of PaWB phytoplasma in an artificial medium has still not been achieved, grafting, DAPI (4′,6-diamidino-2-phenylindole) staining, TEM (transmission electron microscopy) observation and PCR (polymerase chain reaction) detection, have provided sufficient evidence that PaWB phytoplasma is the bacterial pathogen associated with the disease. The PaWB phytoplasma occur as round or elliptical particles with a diameter of 200–820 nm, localized in the sieve tubes of plant phloem when observed with an electron microscope [16]. 

In the paulownia cultivation areas in China, phytoplasmas in the 16SrI-D subgroup were found associated with other plants, such as Japanese maple (*Acer palmatum*) [17], *Kerria japonica* [18], peach (*Prunus persica*) [19], rose balsam (*Impatiens balsamina*) [20], rose (*Rosa rugosa*) [21] and *R. xanthina* [22], *Syringa oblata* and *S. reticulata* [23] and shiny leaf yellow horn (*Xanthoceras sorbifolium*) [24]. Most of them occur nearby or close to the PaWB-diseased paulownia trees.

Paulownias are deciduous tree species in temperate regions. Thus, the distribution of phytoplasmas in paulownia trees varies among different seasons with the temperature [25]. In the northern hemisphere, the phytoplasma concentration in branches and leaves peaks in July and August, when temperatures are high, and the tree is growing vigorously. In the winter, the phytoplasma levels in branches and leaves are very low but still detectable with nested PCR. Changes in phytoplasma content in roots are not as dramatic as those in branches and leaves [26,27].

Genetic diversity exists in the PaWB phytoplasma populations. A population structure study on 142 PaWB phytoplasma strains from 18 provinces in China indicated comparatively high genetic diversity. The phylogenetic analysis was performed through the sequence analysis of 10 housekeeping genes, namely 16S rRNA, *rp*, *fusA*, *secY*, *tuf*, *secA*, *dnaK*, *rpoB*, *pyrG*, *gyrB* and *ipt*. The results showed that the 142 PaWB phytoplasma strains clustered into two main lineages with obvious geographical differentiation [28].

## 4. Transmission of PaWB Phytoplasmas

Phytoplasmas are transmitted by phloem-feeding insect vectors, mainly leafhoppers, planthoppers and psyllids. Transmission experiments have confirmed that stink bugs (such as *Halyomorpha mista* in Japan [29], *H. halys*, and *H. picus* in China, [30]), mirid bugs (such as *Cyrtopeltis tenuis* in Korea [31]) and leafhoppers (such as *Empoasca* sp. in Korea [32] and *Hishimonoides chinensis* in China [33]) are insect vectors of PaWB. In addition to transfer by insect vectors, PaWB phytoplasma can also be transmitted by root cutting, the most popular vegetative propagation material [29]. There are no reports on PaWB phytoplasma transmission by seed [34].

## 5. Symptomatology of PaWB Disease

The most characteristic symptom of PaWB disease is excessive shoot proliferation. Axillary buds break without limit, and shoots and branches grow without restriction (Figure 1). This symptom is commonly called witches’ broom, as it resembles a broom made from bundled twigs that a witch might use to fly, and “Tengu-su” (Tengu’s nest) in Japan, as the numerous short branches resemble the nest of “Tengu”, a mythical, long-nosed goblin who lives in the mountain and flies through the sky [35].

Diseased paulownia trees, regardless the species or geographical area, show the same symptoms. Leaves are smaller and thinner than healthy leaves, and their color fades to yellow. Malformed leaves are often observed on the diseased shoots. The surfaces of diseased leaves are uneven, and the hairs on the underside of the leaves grow sparsely. Reproduction of diseased trees is affected, as phyllody (leafy flower) also occurs. The stigma elongates as a twig, while the sepal, petal and stamen turn into small leaves. Roots of the diseased trees are weak and brownish, easily decay and have little regeneration vigor [10,36]. 

Diseased paulownias begin to show witches’ broom in late spring. The symptoms are not restricted to a specific part of a tree. Shoot proliferation occurs on one or a few branches, then spreads to the adjacent branches until the entire canopy becomes affected (Figure 1). Leaves on the lower part of the diseased shoots fall in late summer or early autumn, while leaves on the upper part of diseased branches remain attached. Axillary buds continue to sprout in late autumn or even in winter. Diseased trees die within a few years after the witches’ broom symptom first appears [10,36].

## 6. Physiological and Anatomic Responses to Phytoplasma Infection

The presence of phytoplasmas alters many physiological indicators in paulownia, such as metabolite content levels and related enzyme activity. Radioisotope tracer assays indicated that the phosphorus/potassium (P/K) ratio, and the total phenolic and chlorogenic acid contents in diseased trees are lower than in healthy ones [37,38]. The protein and vitamin C contents are higher in diseased trees [39,40].

Several indicators of plant stress and disease response are altered in diseased plants. Peroxidase (POD) enzyme activity and malondialdehyde (MDA) contents were higher in diseased *P. fortune* leaves, while superoxide dismutase (SOD) enzyme activity was higher in the healthy leaves [41]. The POD isozyme bands in different organs of diseased *P. tomentosa* show varying degrees of reduction compared to the corresponding organs of healthy trees. Furthermore, the POD isozyme bands in PaWB-diseased trees varied in different seasons [42,43]. 

Phytoplasma infection can also change anatomical structures within paulownia trees. The thickness of the cambium and the diameter of the secondary xylem vessels are reduced in diseased branches. The mesophyll palisade tissue in leaves becomes slim. The amount of hair on the lower page of infected leaves diminishes [44]. The infected small leaves in the diseased trees have blades that are less thick, with reduced and spongy palisade tissues. The length of wood fibers decreases in diseased trees, while the fiber width and cell cavity increase [45].

## 7. Changes in Gene Expression in Phytoplasma-Infected Paulownia

Analyses using simple sequence repeat (SSR) [46,47] and amplified fragment length polymorphism (AFLP) [48,49] molecular markers indicated that infection with phytoplasma did not cause changes in paulownia DNA sequences. The gene expression variation after phytoplasma infection seems to be the key to the induction of disease symptom. The mechanisms underlying gene expression differentiation in phytoplasma-infected paulownia have been studied at the transcriptional, post-transcriptional and translational levels (Table 1).

### 7.1. Transcriptional Responses 

The DNA methylation level within genomes of diseased paulownia seedlings is significantly decreased compared to healthy ones [50]. Treatment with tetracycline can recover health and increase the DNA methylation level of diseased seedlings. Treatment with the DNA methylation agent methyl methane sulfonate (MMS) can also recover the health of diseased in vitro plantlets and increase their DNA methylation level [46]. These clues led to investigations of gene expression patterns, using techniques such as *de novo* transcriptome sequencing and RT-qPCR (retrotranscripted quantitative PCR) on healthy, PaWB-diseased and MMS-treated diseased paulownia plants. Many differentially expressed genes (DEGs) among these samples were identified. Studies using the gene ortholog (GO) functional and Kyoto Encyclopedia of Genes and Genomes (KEGG) pathway databases indicated that these DEGs are involved in various pathways, including phytohormone biosynthesis, photosynthesis and plant–pathogen interaction [51,52,53,54,55,56,57,58].

Third generation sequencing of full-length transcriptomes has also been performed on paulownia, to gauge gene expression responses to phytoplasmas. Combined with metabolome data, a model of the early defense response to PaWB disease was constructed [59].

**Table 1 microorganisms-12-00885-t001:** High-throughput analyses on genomic and gene expression responses to PaWB phytoplasma.

Plant Material	Analysis Methods	Primary Analysis Results	Corresponding Pathways	Key Corresponding Genes	References
*P. fortunei*, healthy, diseased and MMS-treated diseased in vitro plantlets.	AFLP, MSAP and RT-qPCR.	81 methylated fragments and 6 DEGs.			[48]
*P. tomentosa*, healthy, diseased and MMS-treated diseased in vitro plantlets.	AFLP, MSAP and RT-qPCR.	72 methylated fragments and 6 DEGs.			[49]
*Paulownia* spp., healthy and diseased field trees, in vitro plantlets.	Transcriptome sequencing and RT-qPCR.	1271 DEGs in field trees, 1206 DEGs in in vitro plantlets and 19 common KEGG pathways.	Cytokinin biosynthesis, photosynthesis, cell wall biosynthesis and degradation.	Isopentenyl diphosphate isomerase and isopentenyl-transferase.	[51]
*P. tomentosa × P. fortunei*, healthy, diseased and MMS-treated diseased in vitro plantlets.	Transcriptome sequencing and RT-qPCR.	74 significant DEGs and 16 KOGs categories.	General function, posttranslational modification and protein turnover.		[52]
*P. fortunei*, healthy, diseased and MMS-treated diseased in vitro plantlets.	Transcriptome sequencing and RT-qPCR.	1309 DEGs and 83 KEGG pathways.	Plant–pathogen interaction, circadian rhythm, hormone-related.		[53]
*P. fortunei*, healthy, diseased and MMS-treated diseased in vitro plantlets.	Transcriptome sequencing and RT-qPCR.	618 DEGs and 82 KEGG pathways.	Phenylpropanoid metabolism, hormone biosynthesis and signaling, defense and/or pathogenesis and signal transduction.		[54]
*P. tomentosa*, healthy, diseased and MMS-treated diseased in vitro plantlets.	Transcriptome sequencing and RT-qPCR.	2540 DEGs and 119 KEGG pathways.	Folate and fatty acid synthesis and plant hormone signal transduction.		[55]
*P. tomentosa*, *P. fortunei* and *P. tomentosa × P. fortunei*, healthy, diseased and MMS-treated diseased in vitro plantlets.	Transcriptome sequencing and RT-qPCR.	74 DEGs and 12 KEGG pathways.	Phytohormone and alternative splicing.		[56]
*P. tomentosa*, healthy, diseased and MMS-treated diseased in vitro plantlets.	Transcriptome, miRNA and degradome sequencing and RT-qPCR.	902 DEGs, 24 PaWB-related DERs and19 target genes among the DEGs.	Morphological changes, plant defense and plant hormones.	miR156g, miR166c and miR403.	[57]
*P. fortunei*, healthy, diseased and MMS-treated diseased in vitro plantlets.	Transcriptome, miRNA and degradome sequencing and RT-qPCR.	756 DEGs, 67 PaWB-related DERs and 635 target genes among the DEGs.		ARF, WRKY, NAC, MYB and SOD.	[58]
*P. fortunei*, healthy, diseased and MMS-treated diseased in vitro plantlets.	Full-length transcriptome sequencing, LC-ESI-MS/MS metabolism and RT-qPCR.	1561 full length transcripts and 645 metabolites related to PaWB.		SA-binding protein 2, ferulic acid and ethylene-responsive transcription factor RAP2–7 isoform X2.	[59]
*P. tomentosa × P. fortunei*, healthy, diseased and MMS-treated diseased in vitro plantlets.	miRNA and degradome sequencing and RT-qPCR.	33 PaWB-related DERs (13 conserved and 20 novel) and 166 DER target genes.	Plant–pathogen interactions and plant hormone signal transduction, metabolic features.		[60]
*P. fortunei*, healthy, diseased and MMS-treated diseased in vitro plantlets.	miRNA and degradome sequencing and RT-qPCR.	37 PaWB-related DERs (14 conserved and 24 novel) and 147 DER target genes	Transcription, stress response and nitrogen metabolism	miR159-3p, miR169a/b, miR169l, miR171a/b/c/d, miR1-3p, miR17b, miR30-3p and miR32-3p.	[61]
*P. fortunei*, healthy, diseased and MMS-treated diseased in vitro plantlets.	miRNA and degradome sequencing and RT-qPCR.	38 PaWB-related DERs (17 conserved and 20 novel) and 166 target genes of the DERs.		miR160c, miR167, miR171, miR397, miR398a and miR399a/b.	[62]
*P. tomentosa*, *P. fortunei* and *P. tomentosa × P. fortunei*, healthy, diseased and MMS-treated diseased in vitro plantlets.	miRNA and degradome sequencing and RT-qPCR.	76 common DERs (35 miRNA families) from three paulownia species and 196 target genes.	Photosynthesis, plant hormone, plant defense, energy metabolism and material metabolic.	miR156.	[63]
*P. tomentosa*, healthy, diseased and rifampin-treated diseased in vitro plantlets.	High throughput RNA sequencing and RT-qPCR.	1063 PaWB-related mRNAs, 110 PaWB-related lnc RNAs and 12 PaWB-related target genes.			[64]
*P. fortunei*, healthy, and diseased in vitro plantlets.	High throughput RNA sequencing.	2725 lncRNAs and 748 DELs.	Lignin biosynthesis, plant–pathogen interaction and plant hormone signal transduction.		[65]
*P. tomentosa*, healthy, diseased and MMS-treated diseased in vitro plantlets.	High throughput RNA sequencing and RT-qPCR.	3689 lncRNAs, 112 DELs and 51 alternatively spliced target genes.	Reactive oxygen species-induced hypersensitive response and effector-triggered immunity.		[66]
*P. fortunei*, healthy and diseased in vitro plantlets.	High throughput RNA sequencing.	229 lncRNAs, 65 circRNAs and 65 miRNAs, differentially expressed.	Phytohormone biosynthesis, signal transduction, protein processing, amino acid metabolism, chloroplast and defense response.		[67]
*P. fortunei*, healthy, diseased and MMS-treated diseased in vitro plantlets.	m^6^A transcriptome sequencing and m^6^A MeRIP RT-PCR.	315 differential methylated genes.		CLV2, STM, F-box and MSH5.	[68]
*P. tomentosa*, healthy, diseased and MMS-treated diseased in vitro plantlets.	iTRAQ proteome sequencing and RT-qPCR	2051 proteins and 43 PaWB-related DAPs.	Photosynthesis, expression of dwarf symptom, energy production and cell signal pathways.		[69]
*P. fortunei*, healthy, diseased and MMS-treated diseased in vitro plantlets.	iTRAQ proteome sequencing and RT-qPCR	2358 proteins and 36 PaWB-related DAPs.	Carbohydrate and energy metabolism, protein synthesis and degradation and stress resistance.		[70]
*P. fortunei*, healthy, diseased and MMS-treated diseased in vitro plantlets.	iTRAQ proteome sequencing and RT-qPCR.	2969 proteins and 27 PaWB-related DAPs.	Photosynthesis-related, energy-related and ribosome-related.		[71]
*P. tomentosa*, healthy, diseased, MMS-treated healthy and diseased in vitro plantlets.	Quantitative mass spectrometry proteome, acetylome and succinylome.	8963 proteins, 546 PaWB-related acetylated proteins and 5 PaWB-related succinylated proteins.	Protochlorophyllide reductase, RuBisCO and chlorophyll and starch biosynthesis.		[72]
*P. fortunei*, healthy and diseased in vitro plantlets.	Hi-C sequencing RNA-seq.	477 and 510 specific TAD boundaries, 2304 and 3540 specific chromatin loops in healthy and diseased samples and 694 DEGs in common loops.		11 PaWB-closely related genes.	[73]
*P. fortunei*, healthy and diseased in vitro plantlets.	ChIP-seq and ChIP-qPCR.	1821, 1159 and 2727 DMGs marked by H3K4me3, H3K36me3 or H3K9ac and 141 co-modified DMGs.	Metabolic pathways, biosynthesis of secondary metabolites, phenylpropanoid biosynthesis, plant–pathogen interaction and plant hormone signal transduction.		[74]
*P. fortunei*, healthy, diseased and MMS-treated diseased in vitro plantlets.	Chip-seq and RNA-seq.	365, 2244 and 752 PaWB-associated genes with H3K4me3, H3K36me3 and H3K9ac methylation.	Plant–pathogen interaction, plant hormone signal transduction and starch and sucrose metabolism		[75]
*P. fortunei*, healthy and diseased in vitro plantlets.	Bisulfite and transcriptome sequencing, bisulfite-PCR and RT-qPCR.	422,662 DMRs, 27,871 DMR-associated DEGs and 436 genes verified by RNA-seq.	Plant hormone signal transduction, carbon metabolism and starch and sucrose metabolism.	TPR1 and R2R3-MYB	[76]

DERs, differentially expressed RNAs; MS, mass spectrometric; LC-ESI-MS, liquid chromatography electrospray ionization MS; RNA, ribonucleic acid; mRNAs, messenger RNAs; lncRNAs, long noncoding RNAs; DELs, differentially expressed RNAs; circRNAs; circular RNAs; m^6^A, N^6^-methyladenosine; MeRIP, methylated RNA immunoprecipitation; CLV2, CLAVATA2; STM, SHOOT MERISTEMLESS; MSH5, MutS protein homologs 5; iTRAQ, isobaric tags for relative and absolute quantification; DAPs, differentially abundant proteins; RuBisCO, ribulose bisphosphate carboxylase oxygenase; Hi-C, high-throughput chromosome conformation capture; ChIP-seq, chromatin immunoprecipitation sequencing; ChiP-qPCR, chromatin immunoprecipitation quantitative PCR; TADs, topologically associated domains; DMGs, differentially marked genes; DMRs, differentially methylated RNAs; TPR1, TOPLESS RELATED 1.

Mining the paulownia genome data identified some transcription factor (TF) and functional gene families, such as ARF [77], Aux/IAA [78], BTB [79], bZIP [80], CaM/CML [81], GRAS [82], MADS [83], NCED [84], NLR [85], PP2C [86], SERK [87], SPL [88], TCP [89], UBC E2 [90], and WPR [91], that could be affected by phytoplasma colonization. Expression pattern analyses of transcriptome data and using RT-qPCR further showed that some of these gene family members were differentially expressed among healthy, diseased and MMS-treated diseased in vitro paulownia plantlets, indicating that they may be related to the presence of PaWB phytoplasmas (Table 2).

### 7.2. Post Transcript Response 

#### 7.2.1. Noncoding Ribonucleic Acids 

Noncoding RNAs, such as microRNAs (miRNAs), long non-coding RNAs (lncRNAs) and circular RNAs (circRNAs), play important roles in the post-transcriptional regulation of cellular activities. MicroRNAs are small, single-strand, non-coding RNA molecules of 21 to 23 nucleotides. MicroRNAs bind to complementary mRNA sequences, leading to the cleavage or destabilization of mRNA, which negatively impacts the expression of the target gene [92]. LncRNAs are non-coding transcripts of more than 200 nucleotides, while circRNAs are single-strand RNA molecules that form a covalently closed continuous loop. Both are believed to be involved in gene expression regulation, although their functions are not yet clearly defined [93]. 

High-throughput sequencing and degradome analyses have been performed on healthy, diseased and MMS-treated diseased paulownia in vitro plantlets to reveal the functions of noncoding RNAs involved in PaWB. Some differentially expressed miRNAs (DERs) [57,58,60,61,62,63], lncRNAs (DELs) [64,65,66,67] and circRNAs [67], as well as target genes of the DERs/DELs, have been identified and likely play important roles in responding to PaWB infection. A competing endogenous RNA (ceRNA) network that includes these differentially expressed miRNAs, circRNAs, lncRNAs and mRNAs has been constructed [67]. 

#### 7.2.2. Splice Variants

RNA molecules can undergo alternative splicing (AS) to generate multiple mRNA transcript variants from a single gene. AS increases transcriptome plasticity and proteome diversity and is an important post-transcriptional regulatory mechanism. Analysis using high-throughput transcriptome sequencing has identified many alternative splicing variants of paulownia mRNAs [56,58,59,65,66]. The AS frequency in healthy paulownia is higher than in PaWB-diseased samples. Some AS events occur for genes that are differentially expressed between healthy and PaWB-diseased paulownia, implying that splice variants may be related to the paulownia response to PaWB.

#### 7.2.3. Messenger RNA Modification (Epitranscriptome)

Messenger RNA (mRNA) modification is an essential and widespread molecular mechanism underlying key plant developmental process, including embryo development, shoot stem cell fate, floral transition, trichome morphogenesis, leaf initiation and root development [94]. The addition of an N^6^-methyladenosine (m^6^A) is the most common post-transcriptional modification of mRNA. An m^6^A modification is involved in regulating mRNA expression, translation efficiency, alternative splicing and mRNA degradation. The m^6^A transcriptome was sequenced for PaWB-diseased and PaWB-recovered (MMS-treated diseased, 16S rRNA gene-nested PCR-negative) *P. fortunei* plantlets grown in vitro [68]. The results indicated that MMS treatment lowered the m^6^A modification degree in PaWB-recovered paulownia, compared to diseased plantlets. When combined with RNA-seq data, 315 differentially methylated genes (DMGs) were screened from DEGs. Functional enrichment analysis revealed that CLV2 and STM, related to the maintenance of stem cells in shoot apical meristem, were involved in the PaWB–plant interaction. Moreover, alternative splicing was found to be associated with m^6^A modification in F-box and MSH5 genes of MMS-treated diseased plantlets [68].

### 7.3. Translational and Post-Translational Response 

#### 7.3.1. Proteomic Analysis

Proteomic analysis, using isobaric tags for relative and absolute quantification (iTRAQ), can provide accurate, efficient and high-coverage protein quantification. With iTRAQ proteome analysis, differentially abundant proteins (DAPs) among healthy, diseased and MMS-treated diseased paulownia in vitro plantlets were identified. Functional analysis using GO, COG and KEGG databases revealed that proteins related to photosynthesis, energy metabolism and cell signal pathways were most involved in the response of paulownia to phytoplasmas [69,70,71]. 

The correlation between proteome and transcriptome data was poor among the healthy, diseased and MMS-treated diseased paulownia. The number of DAPs was much fewer than the number of DEGs. This suggests that the differences in transcript levels may not be carried through to changes in protein levels. This may be due to the transcriptional/post-transcriptional regulation, translational/post-translational regulation, protein modification or protein–protein interactions [69,70].

#### 7.3.2. Lysine Acetylation and Succinylation

Lysine acetylation and succinylation are post-translational protein modifications that affect protein function and enzyme activity modulating the cellular metabolic status. Proteome, acetylome and succinylome analyses using quantitative mass spectrometry were performed on healthy, diseased, MMS-treated healthy and diseased *P. tomentosa* plantlets grown in vitro. This study revealed that 276 proteins, 546 acetylated proteins and 5 succinylated proteins were regulated, in response to phytoplasma infection. Acetylation of specific sites in protochlorophyllide reductase and RuBisCO are known to modify their activities. The modification of these proteins correlates with the decreased photosynthetic activity and starch accumulation in PaWB-diseased plantlets [72].

### 7.4. Genome Accessibility

#### 7.4.1. Three-Dimensional Chromatin Structure 

The 3-D chromatin conformation in the cell nucleus is closely related to gene expression and epigenetic regulation. High-throughput chromosome conformation capture (Hi-C) sequencing was used to profile the genome-wide chromatin conformation of healthy and PaWB-diseased *P*. *fortunei*. Phytoplasma infection altered the main structural characteristics of active and inactive chromosome regions (also known as A/B compartments), topologically associated domains and chromatin loops in paulownia plants [73]. 

Chromatin immunoprecipitation (ChIP) sequencing and RNA sequencing further showed that the infection with phytoplasmas changed the chromatin structure and transcriptional activity. The modification levels of three histones (H3K4me3/K9ac/K36me3) increased in diseased paulownia. Eleven genes associated with histone marks and located at specific topologically associated domain boundaries, A/B compartment switching and specific loops, were considered to be closely related to the paulownia–phytoplasma interaction [73]. 

#### 7.4.2. Histone Methylation and Acetylation 

Histone methylation and acetylation can influence genome accessibility, at levels ranging from local nucleosome dynamics to the folding of higher chromatin structure. Both methylation and acetylation promote open chromatin conformation and positively influence the genome accessibility [95]. Chromatin immunoprecipitation sequencing (ChIP-seq) was used to profile three histone marks, the methylation of histone H3 at lysine 4 (H3K4me3), the methylation of histone H3 at lysine 36 (H3K36me3) and the acetylation of histone H3 at lysine 9 (H3K9ac), among healthy, diseased and MMS-treated diseased *P. fortunei* plantlets grown in vitro [74,75]. The number of histone modification regions was higher in diseased paulownia than in healthy plants [74]. The combination of RNA-seq data and ChIP-qPCR revealed that histone modification positively activated gene expression. DEGs with differential histone modification profiles were mainly involved in calcium ion signal transduction, abscissic acid signal transduction and ethylene biosynthesis [75].

#### 7.4.3. DNA Methylation

DNA methylation participates in various nuclear processes such as gene expression, DNA repair and recombination. Various studies have indicated that DNA methylation at gene boundaries contributes to a differential gene expression in response to pathogens [95]. Methylation-sensitive amplification polymorphism (MSAP) analysis revealed that the DNA methylation level of healthy plantlets is higher than the PaWB-diseased ones. Treatment with MMS increases DNA methylation levels and induces recovery from PaWB in the plantlets, restoring a healthy morphology. A few genes related to PaWB were discovered through sequencing the MSAP fragments and validated using RT-qPCR [48,49].

A DNA cytosine methylation map was constructed using data from whole genome bisulfite sequencing on healthy and PaWB-diseased *P. fortunei* in vitro plantlets. A total of 422,662 differentially methylated regions (DMRs) were identified. The combination of DNA methylation and transcriptome analyses identified 436 genes that differed in both methylation and transcript levels between healthy and diseased plantlets. Bisulfite-PCR and RT-qPCR validation of the DMR-associated genes suggest that DNA methylation has a repressive effect on gene expression. Two genes, TPR1 and R2R3-MYB, were proposed to be closely related to PaWB [76].

## 8. Genome and Effectors of Paulownia Witches’ Broom Phytoplasma

### 8.1. Paulownia Witches’ Broom Phytoplasma Genome

The complete genome of PaWB phytoplasma comprises 891,641 bp on one circular chromosome with a G + C content of 27.4%. This genome contains 1147 open reading frames, and it is predicted to encode two operons for rRNA genes, as well as 32 tRNA genes. Four potential mobile units (PMUs), ranging from 20 kb to 41 kb, were also identified [8]. Similar with the genome of onion yellows phytoplasma, the PaWB phytoplasma genome carries the genes encoding complete pathways for the metabolism of glycerophospholipid, glycolysis, pyrimidine and folate. One or more of these genes may be absent in the genome of six other phytoplasmas, according to whole genome comparative analysis [8].

### 8.2. Effectors

Seventy-three virulence effectors were predicted in the PaWB phytoplasma genome through analysis using four different bioinformatic tools [8]. The genes for forty PaWB effectors were transcribed in the apical buds of phytoplasma-infected paulownia. The expression of eleven of these effector genes was downregulated after 5 days of treatment with MMS and rifampicin and was undetectable after 20 days of treatment. One of the 11 effectors is an ortholog of the phytoplasma effector SAP54. Expression of SAP54_PaWB_ in *Populus trichocarpa* results in axillary secondary branches along the stem, similar with the formation of witches’ broom symptom. SAP54_PaWB_ interacted with PfSPLa, a member of the Squamosa-promoter binding protein-like (SPL) transcription factor family. The PfSPLa can interact with polyubiquitin and the 26S proteasome non-ATPase regulatory subunit 3 (PfRPN3). This suggests that SAP54_PaWB_ promotes the degradation of PfSPLa in a ubiquitin/26S proteasome-dependent manner to induce shoot proliferation symptoms [8].

A frame diagram of the PaWB molecular pathogenic mechanism is summarized in Figure 2. The genetic information will flow according to DNA to RNA, then to protein. During this progression, epigenetic modifications also influence gene expression at various levels. Phytoplasmas multiply in the sieve cells of the phloem, secrete effector proteins into the neighbor tissue cells and interact with target proteins such as the SPL transcription factors, further influencing the transcription process. These regulation modifications result in normal or PaWB disease morphology. 

## 9. Changes in the Microbiome in Response to Phytoplasma Infection

The plant endophytic microbes have a significant effect on plant nutrient acquisition, abiotic stress tolerance and protection against pathogens [96]. Phytoplasmas are part of the endophyte microbial community and are involved in the symbiotic interaction network. The effect of PaWB phytoplasma infection on the fungal and bacteria communities of paulownia were analyzed by 16S and ITS gene sequencing. The results showed that the abundance and diversity of endophytic fungal species is lower, but the bacterial diversity is higher in the diseased branches and leaves compared to healthy samples. The PaWB phytoplasma infection alters the structure, components and function of the paulownia endophytic microbial community. At the genus level, the abundance of all bacterial genera decreased in the diseased leaf samples except for the ‘*Ca.* Phytoplasma’, including some beneficial bacteria genera such as *Bacillus*, *Pseudomonas*, *Novosphingobium*, *Methylobacterium* and *Sphingomonas* [97,98].

## 10. Paulownia Witches’ Broom Control

### 10.1. Resistance of Paulownia to Phytoplasmas

Resistance to PaWB phytoplasma varies among the paulownia populations. A seven-year continuous observation on different paulownia species and hybrids showed that *P. fortunei*, *P. kawakamii* and *P. tomentosa* are slightly susceptible, each with a disease incidence less than 40% and a disease index lower than 0.15. *P. photeinophylla*, *P. catalpifolia* and *P. australis* are highly susceptible with disease incidences of, or higher than, 60% and disease indices higher than 0.3, while *P. elongata* and *P. fargesii* are moderately susceptible. Among 12 hybrids, *P. fargesii* × *P. fortunei* has the lowest disease incidence (26.7%) and disease index (0.08) [99]. Another observational study on different paulownia species selected one resistant type, *P. tomentosa* var. *tsinlingensis*, which has long and thick leaf hairs that have a long stalk and either a dendritic shape, forked shape, single branch shape or glandular shape with giant cells. All the leaf hairs are sticky, making it difficult for insect vectors to feed on the leaves of this paulownia accession [100].

There have been several successful attempts to induce resistance to PaWB in paulownia. In one study, the selection of disease-free seedlings after grafting with PaWB-diseased materials yielded a paulownia plantation with zero PaWB occurrence [101]. Grafting with diseased material increased the activities of POD, PPO (polyphenol oxidase) and PAL (phenylalanine ammonia-lyase) enzymes in paulownia seedlings and young trees, and thus inducing resistance to PaWB [102]. Resistance to PaWB could also occur through mutations induced by cobalt 60 radiation treatment [103].

### 10.2. Chemical Treatment

Phytoplasmas are sensitive to the antibiotic tetracycline [104]. Treatment with tetracycline, oxytetracycline or rifampicin, by soil drenching and trunk injection, can suppress PaWB symptoms. However, the symptoms may reoccur after the antibiotic treatments are suspended [36,105]. Tetracycline treatment can also be used to recover PaWB diseased plantlets grown in vitro to normal morphology [50]. The DNA methylation agent methyl methanesulfonate (MMS) can also return diseased in vitro plantlets to asymptomatic status, and with no detectable phytoplasma 16Sr RNA gene by nested PCR assay [46].

### 10.3. Field Management

Other management practices include disease-free seedling propagation, insect vector control by pesticides and removal of heavily diseased trees [36]. Treatment of root cuttings with hot water is an effective way to propagate disease-free seedlings. The most efficient treatment conditions are a hot water bath at 46 °C for 15 min [106]. One important cultural practice is the removal of the diseased branches by pruning before leaf fall in the autumn, which prevents phytoplasmas’ movements towards the roots with sap flow, which can reduce symptoms in the next year. On the contrary, the removal of diseased shoots in the spring will make things worse [107]. Thus far, the effective management of PaWB disease on the large scale is still insufficient.

## 11. Perspectives

Paulownia are perennial trees cultivated for wood and ornamental purposes [7]. Unlike other cereal or horticultural crops threatened by phytoplasma diseases [108], there is relatively little input into infected paulownia management. Outside of the necessary irrigation and fertilization when planting, other management measures are rarely used on paulownia plantations. Thus, the paulownia forest community structure is more complex than more uniform farmlands, and the biodiversity is more abundant. As an insect-transmitted disease, it is necessary to understand the interactions among plant hosts, phytoplasmas and insect vectors, especially the feeding behavior of the insect vectors to develop effective control strategies. More efforts should be put into this research area.

Many high-throughput sequencing studies conducted on PaWB provide massive data information for phytoplasma research. The amount of data for paulownia sequencing has reached the terabyte (TB) level, the largest in any phytoplasma-related disease research. These data can be used to shed light also on other phytoplasma disease research. However, the massive amounts of sequencing data can only provide information from a statistical perspective and must rely on bioinformatics analysis tools and logical study design for elaboration. The pathogenic mechanisms underlying molecular alterations discovered from high-throughput data should be verified in detail through biological experimentation. Understanding the mechanisms by which paulownia gene expression responds to phytoplasma effectors is of paramount importance.

The phytoplasma effector SAP54 and homologous have been reported to interact and degrade MADS domain proteins, inducing phyllody symptoms. In the paulownia phytoplasma, the effector SAP54_PaWB_ is verified to interact and degrade the SPL transcription factor protein PfSPLa. The PfSPLa was further annotated as PfSPL4, and it was proven to interact with the promoter region of *PfTCPa* gene [88]. TCP transcript factors play a key role in plant growth regulation and have been reported as targets of phytoplasma effector SAP11 [109], while SPLs are targets of SAP05 [110]. Further research on the downstream influence of SAP54_PaWB_–SPL interaction may reveal novel pathogenic mechanisms of the effectors of this phytoplasma and further help in the clarification of pathogenicity mechanisms in paulownia. 

## Figures and Tables

**Figure 1 microorganisms-12-00885-f001:**
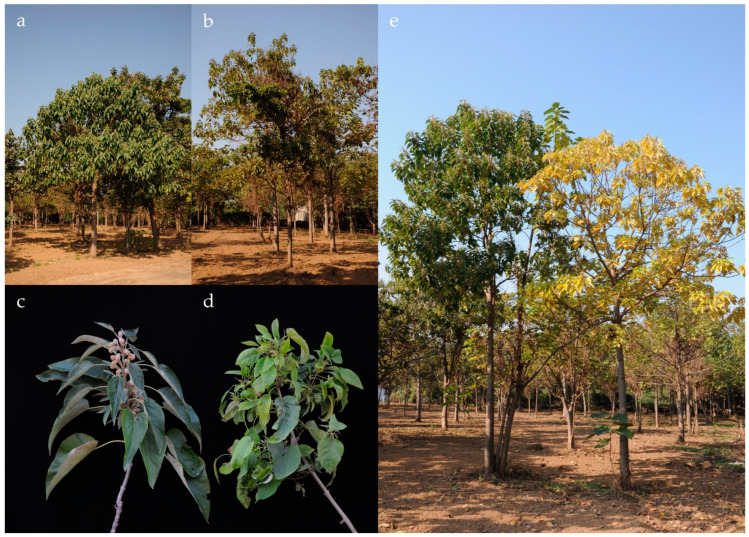
Typical symptoms of PaWB phytoplasma-infected *Paulownia* spp. (**a**) Healthy tree, (**b**) diseased tree, (**c**) healthy branches, (**d**) diseased branches, (**e**) comparison of healthy tree (left) and diseased tree (right).

**Figure 2 microorganisms-12-00885-f002:**
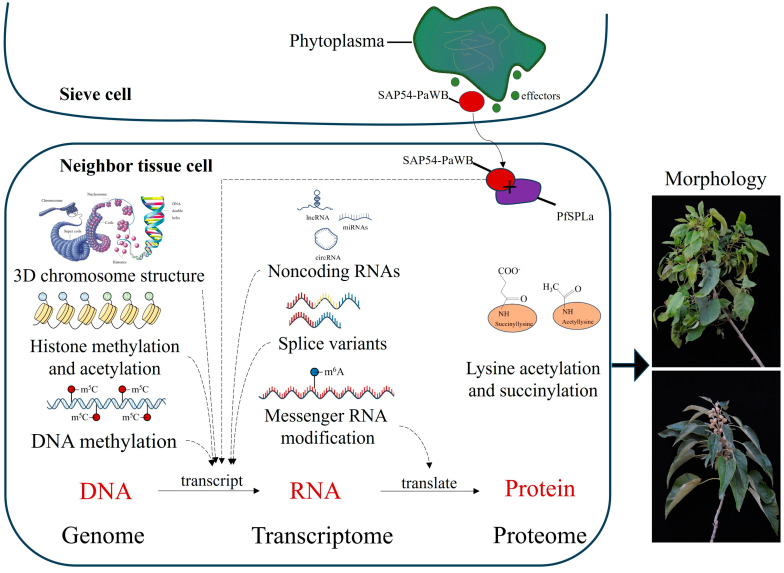
Frame diagram of the PaWB phytoplasma molecular pathogenic mechanism. The phytoplasma multiplies in the sieve cells of the phloem, from which it secret effector proteins to neighbor tissue cells. The frame contains three levels, namely DNA/genome level, RNA/transcriptome level and the protein/proteome level. The mapped gene expression regulation, through genetic information flow according to DNA, RNA and protein, is shown together with the epigenetic modifications and the interaction of the phytoplasma effector SAP54_PaWB_ and the target protein PfSPLa (*Paulownia fortunei* SQUAMOSA-PRO-MOTER BINDING PROTEIN-LIKE a). Dotted lines show potential regulation clues.

**Table 2 microorganisms-12-00885-t002:** Identification of paulownia gene families and PaWB-related paulownia genes.

Gene Family	Data Sets	Family Members	PaWB-Related Genes	References
ARF, auxin response factor	PRJNA624264	33	*PfARF* 18, 21	[77]
Aux/IAA, auxin/indole-3-acetic acid		62	*PfAux/IAA* 13, 33, 45	[78]
BTB, Bric-a-Bric/Tramtrack/Broad complex	PRJNA624264	62	*PfBTB* 3, 12, 14, 16, 19, 36, 44	[79]
bZIP, basic leucine zipper	PRJNA624264	89	*PfbZIP* 46	[80]
CaM/CML, calmodulin and calmodulin-like protein	PRJNA624264	63 (5 CaMs, 58 CMLs)		[81]
GRAS, GAI//RGA//SCL		79	*PfGRAS* 12	[82]
MADS-box	PRJNA794027	89	*PfMADS*3, 57, 87	[83]
NCED, 9-cis-epoxycarotenoid dioxygenase	PRJNA624264	28	*PfNCED*16	[84]
NLR, nucleotide-binding leucine-rich repeat receptors	PRJNA624264	199	*PfNLR* 181	[85]
PP2C, protein phosphatase 2C		91	*PfPP2C* 2, 12, 19, 80	[86]
SERK, somatic embryogenesis receptor-like kinases	PRJNA624264	12	*PfSERK*3, 11	[87]
SPL, SQUAMOSA-PRO-MOTER BINDING PROTEIN-LIKE	PRJNA433928, PRJNA221355, PRJNA289582, SRP060682	23	*PfSPL* 1, 4, 5, 9, 10, 11, 12, 13, 17, 18	[88]
TCP, teosinte branched1, cycloidea, proliferating cell factors		35	*PfTCP* 17, 27	[89]
UBC E2, ubiquitin-conjugating enzyme E2	PRJNA624264	56	*PfUBC* 44, 45, 51	[90]
WPR, WEB1/PMI2-related		16	*PfWEB* 3, *PfWPRb* 2, *PfWPRb* 3, *PfPMI* 2	[91]

## Data Availability

Data are contained within the article.

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
