# Peer review of "Paulownia Witches’ Broom Disease: A Comprehensive Review"

_microorganisms, 2024, doi:10.3390/microorganisms12050885_

Round 1
Reviewer 1 Report
Comments and Suggestions for Authors
In general, the review work is carried out correctly. The structuring of the topics allows for easy and quick reading of the manuscript.
There are some points that I will mention to improve the manuscript.
In the summary section, I suggest making a little more abundant mention of phytoplasma.
In the first paragraph of the introductory section, we elaborate a little more on the pathogenic phytoplasma.
In lines 33-35 (and others in the manuscript). Enter the full name of the microorganisms mentioned. If this is the first time you mention the organism in the manuscript, you must mention it by its full name.
Some names of the microorganisms are not found in italic font.
Lines 124-131. Mention some important examples of the microbiome that are modified (fungi and bacteria)
Finally, I suggest adding a diagram or figure that briefly addresses the pathogenic mechanisms of phytoplasma at the molecular level on paulownia.
Author Response
We thank each reviewer for her/his commitment and valuable time, which contributed to the worthy improvements of our manuscript.
To Reviewer 1
All changes belonging to the comments of reviewer 1 were highlighted in green in the revised version. Other changes are marked by golden letters.
In the summary section, I suggest making a little more abundant mention of phytoplasma.
In the first paragraph of the introductory section, we elaborate a little more on the pathogenic phytoplasmas.
Thank you for the valuable suggestion. We add some more introduction about phytoplasma in the summary and introduction part.
In lines 33-35 (and others in the manuscript). Enter the full name of the microorganisms mentioned. If this is the first time you mention the organism in the manuscript, you must mention it by its full name.
Some names of the microorganisms are not found in italic font.
Thanks a lot! We checked all the names of the microorganisms and made the required corrections.
Lines 124-131. Mention some important examples of the microbiome that are modified (fungi and bacteria)
Agree. We add some more information about the changes of bacteria at genus level here, especially the beneficial bacteria genera.
Finally, I suggest adding a diagram or figure that briefly addresses the pathogenic mechanisms of phytoplasma at the molecular level on paulownia.
Thank for your valuable suggestion. We summarized a PaWB pathogenic mechanism frame diagram on molecular level as Figure 2. For logistic presentation, we alter the order of part 8 and 9 of the original manuscript, and summarize the frame diagram at the end of part 8, genome and effectors of PaWB phytoplasma. For better layout, tables are changed to vertical format.

Reviewer 2 Report
Comments and Suggestions for Authors
The issue addressed in the reviewed paper discusses the problem of Paulownia witches' broom (PaWB) disease, associated with phytoplasma subgroup 16SrI-D. This is an important issue, as it verifies the state of the art on selected aspects of the devastating paulownia disease in East Asia. The topic of the paper is important and in line with the journal's profile.
The authors provided a short historical background and then, in detail, the subject scope, that is: the paulownia witches’ broom phytoplasma. Then we discussed: the transmission of PaWB phytoplasma, the symptomatology of PaWB disease, and reviewed the physiological and anatomical responses to phytoplasma infection.
I have no objections to this part of the study, except for one. I believe that even when doing a review, you have to pose a research question (or more questions) that the literature seeks answers to. Moreover, that question needs to be answered clearly. This is what the structure of a research paper is all about. The result of not having a clear starting question (and thus developing a research problem) is a description rather than a critical review of research in the literature. I recommend that this be corrected and the paper (even just a review), be given a scientific structure (that is, solving a research problem).
Much better developed is the section on changes in gene expression in phytoplasma-infected paulownia. But this section, too, 'misses' the leading point. The readability of the leading issue needs to be improved. So, based on the already well-prepared sections of the paper by the authors, you can go back to the beginning and show the 'road map'. Which is to say? That is, a research scenario, even a review one, which an external reader could repeat on his own and also verify. Although it's 'just' a review, after all, it has its own material and method. But one has to guess which one.
The tables were prepared correctly, as well as the following sections, labeled 8,9 and 10. However, since the leading question of the study was 'lost somewhere', there are no conclusions either (although there are supposedly results). I recommend completing it.
Only then is there space for the important final step, the prospects (Section 11). I also strongly suggest that recommendations for specific, practical, not only general (and not entirely clear) applications of this research shall be provided.
Comments on the Quality of English LanguageThe language of this paper is relatively correct, however some descriptions would benefit from being more concise (please include native speaker verification).
Author Response
We thank each reviewer for her/his commitment and valuable time, which contributed to the worthy improvements of our manuscript.
To Reviewer 2
All changes belonging to the comments of reviewer 2 were highlighted in yellow in the revised version. Other changes are marked in golden letters.
I have no objections to this part of the study, except for one. I believe that even when doing a review, you have to pose a research question (or more questions) that the literature seeks answers to. Moreover, that question needs to be answered clearly. This is what the structure of a research paper is all about. The result of not having a clear starting question (and thus developing a research problem) is a description rather than a critical review of research in the literature. I recommend that this be corrected and the paper (even just a review), be given a scientific structure (that is, solving a research problem).
Thanks for your valuable suggestion. Two questions about PaWB disease study were put forward at the end of the introduction part. One is about the molecular pathogenic mechanism of PaWB, the other is efficient and economically feasible prevention and control methods.
Much better developed is the section on changes in gene expression in phytoplasma-infected paulownia. But this section, too, 'misses' the leading point. The readability of the leading issue needs to be improved. So, based on the already well-prepared sections of the paper by the authors, you can go back to the beginning and show the 'road map'. Which is to say? That is, a research scenario, even a review one, which an external reader could repeat on his own and also verify. Although it's 'just' a review, after all, it has its own material and method. But one has to guess which one.
The tables were prepared correctly, as well as the following sections, labeled 8,9 and 10. However, since the leading question of the study was 'lost somewhere', there are no conclusions either (although there are supposedly results). I recommend completing it.
Thanks a lot. To make a clear “road map” of the PaWB molecular pathogenic study, for logic presentation, we alter the order of part 8 and 9 of the original manuscript, and summarize the frame diagram at the end of part 8, genome and effectors of PaWB phytoplasma.
The changes of bacteria at genus level, especially the beneficial bacteria genera were discussed in part 9, as the most important result in microbiome research.
Only then is there space for the important final step, the prospects (Section 11). I also strongly suggest that recommendations for specific, practical, not only general (and not entirely clear) applications of this research shall be provided.
Thanks a lot, to make the perspectives more specific, the study about PaWB phytoplasma effector SAP54 interact with PfSPL were compared with other interaction between phytoplasma effectors and host plant SPL proteins. Novel clues about phytoplasma effector study were discussed.
